# RECON: MULTIMODAL GRAPHRAG FOR VISUALLY RICH DOCUMENTS WITH INTRA-PAGE REFLECTION AND INTER-PAGE CONNECTION

## ABSTRACT

Multimodal large language models (MLLMs) are widely applied to visual question answering (VQA) for visual documents. However, their ability to comprehend long documents is still constrained by the limited context window. Though recent multimodal retrieval-augmented generation (MMRAG) can assist in retrieving relevant pages to address this challenge, it struggles with questions that require holistic comprehension of the entire document. To cope with this, knowledge graph (KG) that summarizes global knowledge of a document provides an effective solution to enhance the QA performance. However, most existing LLM-based KG-construction methods handle only the language modality; automatically constructing multimodal KGs (MMKGs) for visually rich documents remains largely underexplored. To tackle this issue, we introduce a Multimodal Graph-based RAG approach (namely, RECON), which constructs MMKGs in two stages. (1) Intra-page REflection: it iteratively extracts and reflects both textual and visual entity relations within each page, which is adaptable to the page-content complexity; and (2) Inter-page CONnection: it links multimodal relations across pages to form a coherent global graph. The lack of annotated cross-page global VQA datasets, specifically query-focused visual document summaries (QFVDS), also hinders effective model evaluations. We further build a QFVDS dataset with annotated answers and corresponding supporting facts to enable effective evaluation. Experimental results show that RECON outperforms existing MMRAG approaches on various VQA datasets and QFVDS.

## 1 INTRODUCTION

Recently, MLLMs (OpenAI, 2025; Bai et al., 2025; Comanici et al., 2025; AI, 2025; Team et al., 2025) have achieved strong results on VQA tasks (Kuang et al., 2025). However, their ability to process entire visually rich documents remains limited by context window size and degrades under Context Rot[1] (Hong et al., 2025). MMRAG (Zhang et al., 2025b; Faysse et al., 2025; Yu et al., 2025) mitigates this by retrieving relevant pages, but it fails for questions requiring long-range, document-level understanding.

A promising route is KG construction, where documents are decomposed into entity-centric graphs to connect knowledge across pages. Text-only KGs have been extensively studied Gutiérrez et al. (2025); Ma et al. (2025); Guo et al. (2024); Edge et al. (2024), but building multimodal, visually rich documents automatically introduces new challenges and remains underexplored. Unlike text-only settings, visually rich documents have different page structures, with some pages containing only text and others including further figures or tables with specific layouts, which makes graph construction more demanding. Recently, several works have begun to investigate MMKG-augmented generation (Bu et al., 2025; Lee et al., 2024). These methods highlight the potential of global knowledge, but current approaches either rely on manual graph construction (Lee et al., 2024; Liu et al., 2019) or build graphs dynamically at query time (Bu et al., 2025) with high computation overhead.

---

[1]Context Rot is the degradation of LLM performance with increasing input length, where later tokens are handled less reliably than earlier ones.

This highlights the demand for methods that can automatically construct reusable MMKGs for multimodal reasoning over long-form documents.

Besides constructing MMKGs, evaluation is also a bottleneck. Lacking annotated QA pairs, existing evaluations rely on multi-hop QA benchmarks (Cheng et al., 2024; Tanaka et al., 2023; Chang et al., 2022; Trivedi et al., 2022; Yang et al., 2018), where answers are usually found within two pages. These benchmarks cannot fully capture the difficulty of document-level understanding, which requires information from multiple parts of a document (e.g., "How are major tech companies addressing sustainability?"). This is referred to as a query-focused summarization (QFS) task Edge et al. (2024). In this study, we tackle the QFS for visual documents. While some methods (Guo et al., 2024; Procko & Ochoa, 2024) can automatically generate QFS-style questions, the lack of annotated answers and supporting facts leaves models being evaluated only based on indirect metrics such as fluency and diversity rather than factual correctness Guo et al. (2024); Edge et al. (2024).

To address the above limitations, we introduce RECON, a multimodal graph-based RAG method for visually rich documents, together with a dataset for QFVDS containing annotated answers and supporting facts. Multimodal graph construction should consider text-to-text, text-to-figure, figure-to-figure relations, and the layout of pages. A main difficulty lies in the imbalance across pages; some pages contain only texts while the others include further figures and tables etc., with layouts. To resolve this issue, unlike existing graph construction methods (Gutiérrez et al., 2025; Guo et al., 2024; Edge et al., 2024) relying on a fixed number of extraction rounds, which blocks them from capturing all entities and relations in pages, RECON introduces an adaptive intra-page reflection stage with self-examination. RECON's first stage is an iterative process that extracts and reflects on text-visual entities and relations until no additional information can be retrieved. In the second stage, RECON initializes a graph by joining the same entities across pages. The graph is then refined by linking semantically similar entities to form a coherent global graph. Unlike previous works Gutiérrez et al. (2025); Guo et al. (2024); Edge et al. (2024) which add relations monotonically, a post MLLM examination step is incorporated in the second stage to avoid hallucination of relations. RECON is zero-shot, requiring no training or fine-tuning, making it easy to use in practice.

Our contributions are summarized as follows.
• We advance a step in automatically building MMKGs using MLLMs, an underexplored yet vital direction that can drive progress in graph-based MMRAG for visual documents.
• We introduce a dataset for QFVDS with annotated answers and supporting facts, enabling reliable evaluation of VQA and summarization of globally visual document-level.
• We evaluate RECON on multi-hop QA/VQA benchmarks and QFVDS dataset, covering both text-only and multimodal settings. Experimental results show that RECON outperforms strong MMRAG and graph-based RAG baselines on both text-only and multimodal settings.

## 2 RELATED WORK

We briefly review the related topics of MLLMs, MMRAGs, graph-based RAG, and MMKGs.

**Multimodal Large Language Models.** LLMs such as GPT (OpenAI, 2025), Gemini (Comanici et al., 2025), DeepSeek-R1 (Guo et al., 2025), Qwen (Yang et al., 2025), and others, are significantly advancing the field of natural language processing. Their context windows have also expanded to hundreds of thousands of tokens, yet long-input processing remains brittle: performance degrades as sequences grow, an effect exacerbated by Context Rot (Hong et al., 2025). Building on the advances of LLMs, interest in vision-language interaction has led to the development of MLLMs. Recent models such as Qwen 2.5-VL (Bai et al., 2025), InternVL3.5 (Wang et al., 2025), Llama 4 (AI, 2025), and Gemma 3 (Team et al., 2025) couple vision encoders with LLM backbones and achieve strong results on VQA (Kuang et al., 2025). However, compared to text, visual tokens are far more numerous and can quickly fill up the context window. This leaves MLLMs constrained by context limits and prone to Context Rot, particularly when processing long-form, visually rich documents.

**Multimodal RAG (MMRAG).** To address context window limits and mitigate Context Rot, recent work has turned to MMRAG. The idea is to retrieve only the relevant document pages, reducing input length while preserving structural information. DSE (Ma et al., 2024) follows this approach by encoding document screenshots directly, combining visual layout, text, and images into a unified vector representation. ColPaLi (Faysse et al., 2025) improves on this with multi-vector em-

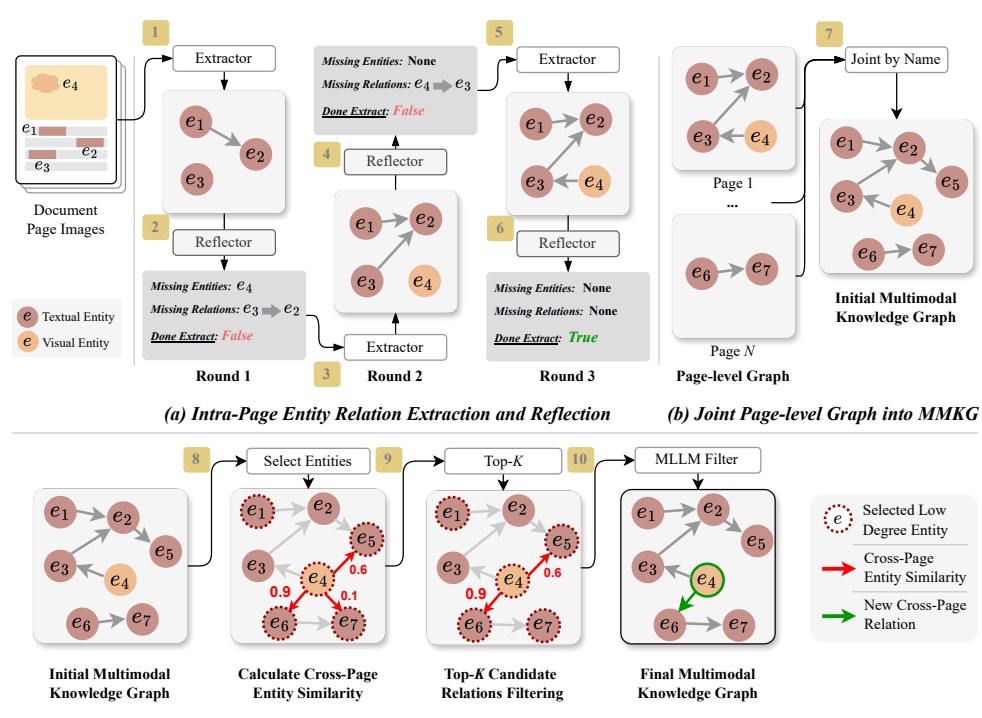

Figure 1: Overview of RECON for MMKG construction, consisting of two stages. (a) Intra-Page Reflection: Each page image is first processed in parallel by an extractor (Step 1) to extract textual and visual entities with their relations. A reflector (Step 2) checks for missing entities and relations, and if any are found, the page undergoes another extraction-reflection round (Steps 3-5). This loop continues until no further entities or relations are missing (Step 6). (b) The refined page-level graphs are then merged into an initial MMKG (Step 7). (c) Inter-Page Connection: To capture cross-page relations more efficiently, we first select low-degree entities using a knee threshold (Step 8). We then compute their semantic similarity with entities from other pages and select the top-$k$ candidates (Step 9). These candidates are passed to an MLLM filter, which validates whether true relations exist (Step 10). Confirmed relations are then added back into the graph, yielding a final MMKG.

beddings of document images, capturing fine-grained visual cues. Its variant, ColQwen, replaces PaLI-Gemma (Beyer et al., 2024) with Qwen2.5-VL (Bai et al., 2025), yielding stronger retrieval performance. Moving beyond retrieval, VisRAG (Yu et al., 2025) integrates an MLLM into the full RAG pipeline, embedding document images for retrieval and reusing them in answer generation.

The above methods excel in text-to-image retrieval but fail to solve tasks involving a mixture of single-modality (e.g., image-to-image) and fused-modality (text+image-to-text+image) retrieval. GME (Zhang et al., 2025b) tackles this by introducing a unified embedding model that encodes diverse modality combinations and enables flexible retrieval within a shared representation space.

Yet current MMRAG methods remain limited to page-level retrieval and overlook document-level structure, which is essential for answering questions that require reasoning across hundreds of pages.

**Graph-based RAG.** A direction for document-level reasoning is KG construction, where documents are decomposed into entity-centric graphs that connect knowledge across pages. Knowledge graph-augmented generation (Zhang et al., 2025a) methods such as SubgraphRAG (Li et al., 2025), G-Retriever (He et al., 2024), and Gao et al. (2022) improve retrieval efficiency through subgraph selection or learning-to-rank, while ToG-2 (Ma et al., 2025) introduces a hybrid RAG method that alternates between dense retrieval and graph reasoning. However, these approaches rely on manually curated KGs, which are costly to construct and limited in coverage. To overcome these limitations, GraphRAG (Edge et al., 2024) constructs KGs directly from raw text using LLMs and organizes them with hierarchical community detection (Traag et al., 2019), enabling document-level reasoning

but at high computational cost. LightRAG (Guo et al., 2024) improves scalability by discarding community summaries and instead leveraging dense retrieval of entities and relations. HippoRAG 2 (Gutiérrez et al., 2025) enhances graph-based RAG by combining Personalized PageRank with deeper passage integration, achieving stronger performance on single and multi-hop QA tasks.

Despite these advances, graph-based RAG remains largely limited to text-only and cannot capture multimodal signals such as images, tables, and layouts, which are crucial in visually rich documents.

**Multimodal Knowledge Graphs (MMKGs).** MMKGs extend conventional KGs by associating entities with aligned visual information (Liu et al., 2019; Zhang et al., 2023). Early work (Liu et al., 2019) was manually constructed by linking overlapping entities and augmenting them with web-crawled images and numeric literals. MMKGs have shown utility across tasks such as KG completion (Mousselly-Sergieh et al., 2018; Xie et al., 2017), recommendation (Sun et al., 2020), and image captioning (Zhao & Wu, 2023). More recently, several studies have investigated MMKG-augmented generation (Bu et al., 2025; Lee et al., 2024). Lee et al. (2024) leverage manually curated MMKGs to guide LLMs in answer generation, but these graphs are expensive to build and domain-specific. Bu et al. (2025) proposes a query-driven MMKG that employs MLLMs to dynamically construct MMKGs at query time, improving flexibility but incurring high computational overhead since a new graph must be generated for each query.

These methods highlight the potential of MMKGs but are constrained by manual construction and costly query-time generation. To overcome these limitations, we propose RECON, a reusable MMKG method for robust reasoning over long-form, visually rich documents.

## 3 Methodology

In this section, we present RECON, covering the construction process of MMKG, graph indexing and retrieval mechanisms, the answer generation pipeline, and the QFVDS construction methods.

### 3.1 MMKG Construction

We define the MMKG as $\mathcal{G}$, representing a graph of entities and relations. Given a document with $N$ pages, we first extract the rendered page image $\mathrm{P}_i$ (which captures the layout of the page) from the document and feed it into our graph construction pipeline.

**Intra-Page Extraction and Reflection.** As illustrated in Figure 1(a), each page is first processed in parallel by an extractor $G(\cdot)$, which employs an MLLM guided by a task-specific prompt to identify textual and visual entities along with their relations. Given a page input $\mathrm{P}_i$, the extractor produces an initial set of entities and relations $(\mathrm{E}_i^{(0)}, \mathrm{R}_i^{(0)}) = G(\mathrm{P}_i)$. Here, entities may originate from textual content or from visual objects depicted in the figures of the page. Each entity is represented by a name, a type (e.g., person, organization), its modality (textual or visual), and a short description. Relations are defined by their source and target entities, a description, and representative keywords.

Since the extractor may omit entities or relations, we apply a reflector $D(\cdot)$, an MLLM with a tailored prompt, to re-examine both the original page $\mathrm{P}_i$ and the extracted graph $(\mathrm{E}_i^{(0)}, \mathrm{R}_i^{(0)})$. The reflector outputs additional candidates $(\hat{\mathrm{E}}_i^{(0)}, \hat{\mathrm{R}}_i^{(0)}) = D(\mathrm{P}_i, \mathrm{E}_i^{(0)}, \mathrm{R}_i^{(0)})$, which complement the extractor results. If the candidate set is not empty, the page together with the current graph $(\mathrm{E}_i^{(0)}, \mathrm{R}_i^{(0)})$ and the candidate additions $(\hat{\mathrm{E}}_i^{(0)}, \hat{\mathrm{R}}_i^{(0)})$ is passed into another extractor-reflector round. This iterative process continues until the reflector yields no further additions, resulting in the final refined set $(\mathrm{E}_i^*, \mathrm{R}_i^*)$, where the superscript * denotes the termination state after all extractor-reflector iterations.

As shown in Figure 1(b), after processing all pages, the outputs $\{(\mathrm{E}_i^*, \mathrm{R}_i^*)\}_{i=1}^N$ are joined into a unified MMKG $\mathcal{G}$. Following the merging strategy of LightRAG, entity nodes with the same name are merged, and relation edges with the same source and target are consolidated into a single edge. To handle cases where the same entity or relation has multiple descriptions, an MLLM merges them into a single sentence and unifies keywords from repeated occurrences.

**Inter-Page Connection.** Beyond merging same-name entities across pages, we further capture cross-page relations by linking entities into a more coherent MMKG, as illustrated in Figure 1(c). For efficiency, instead of considering all entities, we focus on those with few connections by filtering

out already well-connected nodes. Since entity degrees often follow a long-tailed distribution (Mirtaheri et al., 2024), we define low-degree entities as those below a threshold automatically determined by a knee detection method (Satopaa et al., 2011).

Next, for each low-degree entity, we construct an entity-centric query to search for potential cross-page connections. Given an entity's name and description, the query is formulated as *"Find entities associated with name: {**name**}, description: {**description**}."* We then use this query to measure semantic similarity against all other entities in the graph (detailed in Section 3.2), excluding those from the same page or already connected. From these comparisons, we select the top-$k$ most semantically similar candidates. While this step helps surface plausible cross-page links, relying solely on semantic similarity risks introducing hallucinatory edges. To alleviate this, each low-degree entity and its candidate set are passed to an MLLM filter, which validates whether genuine relations exist. Confirmed pairs are then added to $\mathcal{G}$, thereby enriching the graph with reliable cross-page links.

## 3.2 MMKG Retrieval and Answer Generation

RECON adopts a unified retrieval method that integrates graph structure and images within a shared embedding space. Specifically, we employ GME (Zhang et al., 2025b), a multimodal encoder that embeds both text and images into a common vector space.

**Indexing and Retrieval.** We build indexes for three contents: entities, relations, and page images. To represent an entity, we join its name with its description into a single sentence and encode it with GME. Relations are encoded in the same way, using the names of the source and target entities, relation keywords, and their description. For page images, we directly embed the image using GME.

Given a user query, we first use the MLLM to extract two types of phrases: (1) entity mentions in the query, and (2) topics that the query discusses. These phrases are then embedded with GME and compared to the entity store using dot-product similarity, returning the top-$u$ most relevant entities. In parallel, the same embedded phrases are also used to search the relation store, retrieving the top-$u$ most relevant relations along with their source and target entities. To provide additional context, each retrieved entity is further expanded with its one-hop neighbors from $\mathcal{G}$.

Complementary to entity and relation retrieval, we perform page retrieval to capture document layout cues that the MMKG might miss. The query sentence is embedded by GME and compared with page image embeddings, retrieving the top-$m$ relevant pages.

**Answer Generation.** Our goal is to leverage the cross-page knowledge captured by the MMKG, together with the local details from document pages to answer user questions. A straightforward approach would be to feed the retrieved entities, relations, and pages directly into the MLLM. However, this often introduces modality bias (Chen et al., 2024), where the model prioritizes one modality (typically text). To cope with this, we use a two-stage process. First, the MLLM creates two answers in parallel: one from the retrieved entities and relations, and one from the retrieved page images. Then, it combines these two intermediate answers to produce the final response.

Full prompt formats for each stage are provided in Appendix A.5.

## 3.3 QFVDS Dataset Construction

Constructing QFVDS follows a two-stage pipeline. First, we extract atomic facts from documents and organize them into semantically coherent topics. Second, we use these topics to generate queries, answers, and facts, followed by human validation to ensure quality.

**Fact Extraction and Topic Clustering.** We collected four types of visually rich documents: (1) environmental reports from tech companies, (2) a world history textbook, (3) DLCV course slides on deep learning for computer vision, and (4) picture books, most of which contain little or no text. For each document set, we use an MLLM to extract facts from every page. These facts are derived from both textual and visual elements and written to be fully self-contained. We embed all extracted facts and cluster them into coherent topics using HDBSCAN (McInnes et al., 2017). Each cluster is then summarized by an MLLM into a topic summary describing what the grouped facts are about.

**QA Generation and Human Validation.** To create queries that require long-range reasoning, we randomly sample topics at different scales (5, 10, 15, and 20 topics per sample). Each setting

is repeated 50 times to form diverse subsets of facts. The sampled topics, their summaries, and underlying facts are then given to an MLLM to generate a question and its answer.

We then conduct a manual review to filter out duplicates and unreasonable QA pairs. The final dataset contains 138 QA pairs for environmental reports, 187 for world history, 172 for DLCV slides, and 127 for picture books. Further details of QFVDS are provided in Appendix A.1.

## 4 EXPERIMENTS

In this section, we outline the experimental setups and present the results for our RECON method.

### 4.1 DATASETS

We evaluate RECON on two types of datasets: existing multi-hop QA benchmarks and our proposed QFVDS dataset.

**Multi-hop QA Benchmarks.** For text-only QA, we use HotpotQA (Yang et al., 2018), MuSiQue (Trivedi et al., 2022), and MultiHopQA (Cheng et al., 2024). Following HippoRAG2 (Gutiérrez et al., 2025), we randomly sample 1,000 queries from each dataset. For multimodal QA, we include SlideVQA (Tanaka et al., 2023) and WebQA (Chang et al., 2022). From SlideVQA, we use the multi-hop subset (363 queries), and from WebQA we randomly sample 1,000 multi-hop queries. These benchmarks test multi-hop reasoning over a few passages or pages, but they may not capture the full difficulty of reasoning across long documents.

**QFVDS Dataset.** To evaluate document-level VQA, we introduce QFVDS, built from four types of visually rich documents (detailed in Section 3.3): (1) World History: a world history textbook (788 pages, 187 QA pairs), (2) Environmental Reports: sustainability reports from Google, Apple, Meta, and Nvidia (422 pages, 138 QA pairs), (3) DLCV Slides: lecture slides on deep learning for computer vision (1,984 pages, 172 QA pairs), and (4) Picture Books: 10 books (247 pages, 127 QA pairs), most with little or no text. In total, QFVDS provides 624 QA pairs with annotated answers and supporting facts. Unlike multi-hop QA benchmarks, QFVDS is designed to test long-range reasoning across entire documents.

### 4.2 BASELINES AND EVALUATION METRICS

**Baselines.** We compare RECON against both embedding-based RAG and graph-based RAG. As a lower bound, we include a "No Documents" baseline, where the MLLM answers questions without any external content, testing whether internal knowledge alone is sufficient. For embedding-based retrieval, we use NaiveRAG with dense retrieval (OpenAI's *text-embedding-3-small*) in text-only settings, and VisRAG (Yu et al., 2025), ColQwen (Faysse et al., 2025), and GME (Zhang et al., 2025b) for multimodal settings. For graph-based RAG, we evaluate GraphRAG (Edge et al., 2024), LightRAG (Guo et al., 2024), and HippoRAG2 (Gutiérrez et al., 2025).

**Metrics for Multi-hop QA.** We evaluate performance by comparing the generated answers against ground truth answers. Specifically, an LLM is used to judge whether the generated answer aligns semantically with the reference answer. Accuracy is then computed based on the proportion of correct matches.

**Metrics for QFVDS.** We adopt the pairwise evaluation method used in GraphRAG (Edge et al., 2024). Unlike previous works, QFVDS includes both annotated answers and supporting facts, which allows us to evaluate with two metrics: *Semantic Similarity*, measuring how well the generated answer matches the reference, and *Factualness*, assessing whether the predicted supporting facts correctly match and fully cover the annotated ones.

To mitigate positional bias in pairwise evaluation, we randomize answer order. Specifically, we show the LLM two answers without revealing which method they come from (e.g., RECON vs. Baseline), and ask which is better. Then, we swap the order (Baseline vs. RECON) and ask again, so that each answer appears first once. If the LLM selects the first-position answer in both rounds, we treat it as a tie, as this may reflect position bias rather than a genuine preference.

The detailed prompts are shown in Appendix A.5.

| Method | Text-Only | | | | Multimodal | | |
|---|---|---|---|---|---|---|---|
| | HotpotQA | MuSiQue | MultiHop | *Avg.* | SlideVQA | WebQA | *Avg.* |
| *Without Retrieval* | | | | | | | |
| No Documents | 37.40 | 16.40 | 38.20 | 30.67 | 9.92 | 37.30 | 23.61 |
| *Embedding RAG* | | | | | | | |
| NaiveRAG | 69.50 | 43.30 | 48.10 | 53.63 | 28.93 | 25.80 | 27.37 |
| VisRAG | – | – | – | – | 60.61 | 56.30 | 58.46 |
| ColQwen | – | – | – | – | 64.19 | 56.70 | 60.45 |
| GME | – | – | – | – | 65.67 | 56.10 | 60.89 |
| *Graph-based RAG* | | | | | | | |
| GraphRAG | 44.50 | 33.80 | 46.30 | 41.53 | 19.56 | 26.80 | 23.18 |
| LightRAG | 60.00 | 29.70 | 60.50 | 50.07 | 17.63 | 13.80 | 15.72 |
| HippoRAG2 | 80.60 | 46.90 | **73.10** | 66.87 | 35.81 | 25.50 | 30.66 |
| RECON | **81.30** | **47.00** | 72.50 | **66.93** | 66.12 | 60.50 | 63.31 |

Table 1: Accuracy (%) on multi-hop QA across text-only and multimodal settings. "—" denotes not evaluated in the text-only setting because these models require visual inputs. "No Documents" means the MLLM answers questions without any external content.

| Baseline | Metric | World History Book | | | | Picture Books | | | | Environmental Reports | | | | DLCV Slides | | | |
|---|---|---|---|---|---|---|---|---|---|---|---|---|---|---|---|---|---|
| | | RECON > | Baseline > | Tie | Δ ↑ | RECON > | Baseline > | Tie | Δ ↑ | RECON > | Baseline > | Tie | Δ ↑ | RECON > | Baseline > | Tie | Δ ↑ |
| *Without Retrieval* | | | | | | | | | | | | | | | | | |
| No Document | SemSim | **82.30** | 7.00 | 10.80 | 75.30 | **81.00** | 8.70 | 10.30 | 72.20 | **82.60** | 4.30 | 13.00 | 78.30 | **95.90** | 0.60 | 3.50 | 95.30 |
| | Faithful | **71.00** | 12.90 | 16.10 | 58.10 | **47.60** | 18.30 | 34.10 | 29.40 | **37.70** | 16.70 | 45.70 | 21.00 | **82.40** | 2.90 | 14.70 | 79.40 |
| *Embedding RAG* | | | | | | | | | | | | | | | | | |
| NaiveRAG | SemSim | **29.90** | 26.20 | 43.90 | 3.70 | **48.40** | 16.70 | 34.90 | 31.70 | **31.40** | 20.40 | 48.20 | 10.90 | **37.10** | 22.80 | 40.10 | 14.40 |
| | Faithful | **28.30** | 27.80 | 43.90 | 0.50 | **46.80** | 16.70 | 36.50 | 30.20 | **32.80** | 24.10 | 43.10 | 8.80 | **36.50** | 22.20 | 41.30 | 14.40 |
| VisRAG | SemSim | **45.70** | 25.00 | 29.30 | 20.70 | **43.70** | 31.00 | 25.40 | 12.70 | **54.10** | 13.30 | 32.60 | 40.70 | **50.60** | 17.10 | 32.40 | 33.50 |
| | Faithful | **42.90** | 26.60 | 30.40 | 16.30 | **39.70** | 34.10 | 26.20 | 5.60 | **48.90** | 17.80 | 33.30 | 31.10 | **43.50** | 21.20 | 35.30 | 22.40 |
| ColQwen | SemSim | **32.10** | 23.50 | 44.40 | 8.60 | **50.40** | 14.40 | 35.20 | 36.00 | **29.20** | 19.70 | 51.10 | 9.50 | **38.30** | 25.10 | 36.50 | 13.20 |
| | Faithful | **31.00** | 25.10 | 43.90 | 5.90 | **50.40** | 14.40 | 35.20 | 36.00 | **29.90** | 23.40 | 46.70 | 6.60 | **37.70** | 23.40 | 38.90 | 14.40 |
| GME | SemSim | **47.50** | 28.70 | 23.80 | 18.80 | **45.60** | 20.00 | 34.40 | 25.60 | **38.20** | 19.10 | 42.60 | 19.10 | **52.00** | 12.30 | 35.70 | 39.80 |
| | Faithful | **45.30** | 29.30 | 25.40 | 16.00 | **37.60** | 23.20 | 39.20 | 14.40 | **36.00** | 22.10 | 41.90 | 14.00 | **47.40** | 18.10 | 34.50 | 29.20 |
| *Graph-based RAG* | | | | | | | | | | | | | | | | | |
| GraphRAG | SemSim | **59.10** | 10.20 | 30.60 | 48.90 | **60.50** | 12.10 | 27.40 | 48.40 | **39.60** | 16.40 | 44.00 | 23.10 | **49.70** | 17.40 | 32.90 | 32.30 |
| | Faithful | **59.70** | 10.20 | 30.10 | 49.50 | **56.50** | 13.70 | 29.80 | 42.70 | **39.60** | 15.70 | 44.80 | 23.90 | **49.70** | 16.80 | 33.50 | 32.90 |
| LightRAG | SemSim | **34.90** | 30.60 | 34.40 | 4.30 | **63.20** | 8.00 | 28.80 | 55.20 | **23.90** | 21.70 | 54.30 | 2.20 | **45.80** | 10.10 | 44.00 | 35.70 |
| | Faithful | **33.90** | 30.60 | 35.50 | 3.20 | **60.00** | 8.00 | 32.00 | 52.00 | **25.40** | 24.60 | 50.00 | 0.70 | **41.10** | 13.70 | 45.20 | 27.40 |
| HippoRAG2 | SemSim | **47.80** | 22.30 | 29.90 | 25.50 | **59.70** | 5.00 | 35.30 | 54.60 | **37.80** | 19.30 | 51.10 | 18.50 | **49.40** | 15.90 | 34.70 | 33.50 |
| | Faithful | **45.10** | 22.80 | 32.10 | 22.30 | **52.10** | 10.10 | 37.80 | 42.00 | **34.10** | 23.00 | 43.00 | 11.10 | **43.50** | 17.60 | 38.80 | 25.90 |
| *Avg.* | SemSim | **47.40** | 21.70 | 30.90 | 25.70 | **56.50** | 14.50 | 29.00 | 42.10 | **42.10** | 16.80 | 41.10 | 25.30 | **52.40** | 15.10 | 32.50 | 37.20 |
| | Faithful | **44.70** | 23.20 | 32.20 | 21.50 | **48.80** | 17.30 | 33.90 | 31.50 | **35.50** | 20.90 | 43.60 | 14.60 | **47.70** | 17.00 | 35.30 | 30.70 |

Table 2: Win Rate (%) of RECON vs. each baseline on QFVDS dataset. For each domain, we report RECON> (win), Baseline> (win), Tie, and Δ (RECON − Baseline) under two metrics: SemSim (semantic similarity) and Faithful (faithfulness). "No Documents" means the MLLM answers questions without any external content.

## 4.3 IMPLEMENTATION DETAILS

We standardize the implementation of all RAG methods to ensure fair comparison. For all graph construction and answer generation, we use *GPT-4o-mini*, consistent with the original setups of GraphRAG and LightRAG. For evaluation and QFVDS dataset construction, we adopt *o4-mini* for more robust judgments. The generation temperature is fixed to 0 across all tasks to reduce output variance. For text-only QA settings, documents are segmented into 1,200-token chunks with a 100-token overlap, and prompts are kept consistent across methods to avoid prompt-induced variability.

For embedding-based RAG, we use OpenAI's *text-embedding-3-small* for NaiveRAG, *visrag-ret* (Yu et al., 2025) for VisRAG, *colqwen2.5-v0.2* (Faysse et al., 2025) for ColQwen, and *gme-qwen2-vl-7b* (Zhang et al., 2025b) for GME. For graph-based RAG, GraphRAG and LightRAG both use *text-embedding-3-small*, HippoRAG2 uses *nv-embed-v2* (Lee et al., 2025), following its own setup.

For RECON, we use *gme-qwen2-vl-7b* (Zhang et al., 2025b) in multimodal QA settings. For text-only QA settings, we use *text-embedding-3-small* to ensure fair comparison with other text-only baselines. During MMKG construction, we limit the extractor-reflector loop to a maximum of 5 rounds and set $k = 10$ for the number of inter-page connection candidates. For retrieval, we retrieve up to $u = 60$ entities and relations and $m = 6$ pages.

| Ablation | Metric | Environmental Reports | | | | Picture Books | | | |
|---|---|---|---|---|---|---|---|---|---|
| | | RECON > | Ablation > | Tie | Δ ↑ | RECON > | Ablation > | Tie | Δ ↑ |
| w/o Intra-Page Reflection | SemSim | **30.40** | 16.30 | 53.30 | 14.10 | **43.20** | 25.60 | 31.20 | 17.60 |
| | Faithful | **30.40** | 20.00 | 49.60 | 10.40 | **43.20** | 26.40 | 30.40 | 16.80 |
| w/o Inter-Page Connection | SemSim | **28.90** | 25.20 | 45.90 | 3.70 | **35.50** | 28.20 | 36.30 | 7.30 |
| | Faithful | **28.90** | 26.70 | 44.40 | 2.20 | **33.90** | 27.40 | 38.70 | 6.50 |
| w/o Textual Entity | SemSim | **37.60** | 17.20 | 45.20 | 20.40 | **37.30** | 30.20 | 32.50 | 7.10 |
| | Faithful | **38.40** | 17.20 | 44.50 | 21.20 | **37.30** | 29.40 | 33.30 | 7.90 |
| w/o Visual Entity | SemSim | **32.10** | 21.20 | 46.70 | 10.90 | **48.40** | 21.00 | 30.60 | 27.40 |
| | Faithful | **31.40** | 21.90 | 46.70 | 9.50 | **47.60** | 21.00 | 31.50 | 26.60 |

Table 3: Ablation study (pairwise win rates, %). Larger $\Delta$ (RECON − Ablation) values imply greater importance of the removed component.

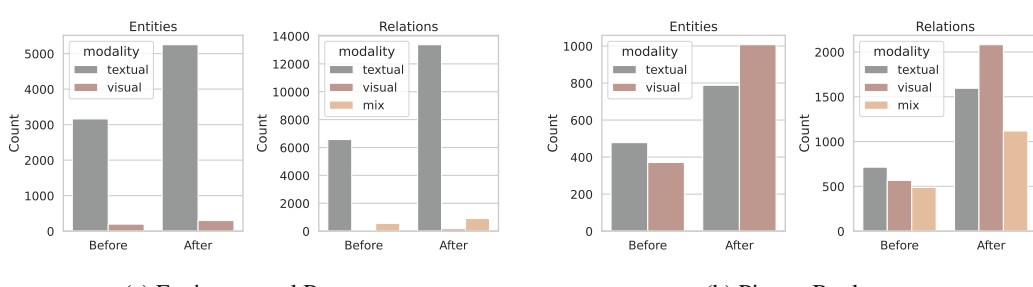

(a) Environmental Reports.   (b) Picture Books.

Figure 2: Entity and relation counts with and without intra-page reflection for two datasets. Here, mix relations denote cross-modal links that connect textual and visual entities.

For baselines without multimodal support (NaiveRAG, GraphRAG, LightRAG, and HippoRAG2), we use the MinerU toolkit (Wang et al., 2024) to extract text from visual documents and process it using the same pipeline as the text-only settings described above.

## 4.4 MAIN RESULTS

**Multi-hop QA.** Table 1 reports results on both text-only and multimodal multi-hop QA benchmarks. The "*No Documents*" baseline reveals that MLLMs alone struggle to answer multi-hop questions, with average accuracy dropping to 30.67% on text-only and 23.61% on multimodal datasets. This underscores the importance of external document grounding.

On text-only datasets, RECON performs on par with HippoRAG2, achieving an average accuracy of 66.93%. It obtains the best results on HotpotQA (81.3%) and MuSiQue (47.0%), while remaining competitive on MultiHopQA (72.5%). These results indicate that the adaptive reflection mechanism can also be beneficial in text-only settings.

On multimodal datasets, RECON shows substantial gains. It improves accuracy by more than 30 percentage points over HippoRAG2 (63.31% vs. 30.66%) and outperforms the evaluated MMRAG baselines. On SlideVQA, the margin is smaller because many questions contain explicit keywords that appear directly in the slides, which makes MMRAG methods particularly effective. Nevertheless, RECON remains competitive on SlideVQA and achieves strong results on WebQA. Overall, these results highlight the benefit of MMKG, which enables RECON to reason across modality and pages.

**QFVDS.** Table 2 reports win-rate comparisons between RECON and all baselines across four domains of QFVDS. First, the "*No Documents*" baseline performs poorly, showing that MLLMs cannot rely solely on internal knowledge to answer document-level questions.

Among embedding-based methods, RECON consistently outperforms NaiveRAG, VisRAG, ColQwen, and GME across all domains. Since these methods mainly retrieve page-level content, they tend to achieve relatively higher faithfulness. However, their reliance on local retrieval limits answer diversity, which results in lower semantic similarity when the task requires integrating information beyond a single page.

Graph-based methods (GraphRAG, LightRAG, HippoRAG2) narrow the gap in some domains, but RECON still achieves higher win rates overall. For example, on Picture Books and DLCV Slides, RECON shows large improvements in both semantic similarity (up to +42.1% and +37.2%) and faithfulness (up to +31.5% and +30.7%, respectively). These gains highlight that RECON's MMKG construction is particularly effective in settings where information is spread across long documents (e.g., lecture slides with many pages) or where textual content is limited and visual elements such as figures and tables play a central role.

## 4.5 ABLATION STUDY

Table 3 presents ablations on four components of RECON: intra-page reflection, inter-page connection, textual entities, and visual entities.

Removing the intra-page reflection leads to a large degradation in both domains. This confirms that iterative extraction is crucial for uncovering missing entities and relations. Excluding the inter-page connection produces moderate yet consistent drops, suggesting that cross-page linking is necessary to integrate evidence that would otherwise remain fragmented across distant pages.

For modality-specific ablations, removing textual entities from MMKG results in larger declines on Environmental Reports, where content is primarily conveyed through text and tables. Conversely, removing visual entities causes the most severe degradation on Picture Books, which rely heavily on images. These results demonstrate that RECON benefits from both textual and visual nodes, with their relative importance determined by the modality balance of the underlying documents.

## 4.6 DETAILED ANALYSIS

**Entity and Relation Gains.** Figure 2 shows how iterative reflection increases the number of extracted entities and relations across modalities. For Environmental Reports, the reflection process substantially boosts both textual entities and relations, with additional gains also observed for visual and mixed relations. For Picture Books, the effect is more pronounced on visual nodes and relations, where reflection nearly doubles the extracted counts. This highlights the importance of multiple rounds in capturing implicit or small visual details that are often missed initially.

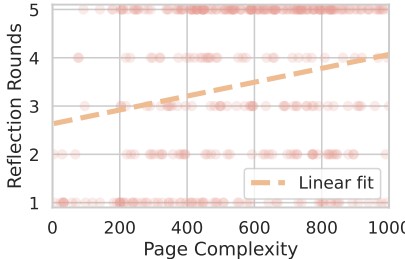

Figure 3: Reflection rounds and Page Complexity on Environmental Reports.

**Reflection Rounds.** Figure 3 shows the association between page complexity and the number of reflection rounds. Page complexity is defined as a combination of word count and the number of tables and figures, with counts obtained using the MinerU document parsing tool (Wang et al., 2024). The analysis reveals a positive Pearson correlation ($r=0.219$, $p=2.49 \times 10^{-5}$), indicating that more complex pages generally require additional reflection rounds. While the correlation is moderate, the trend highlights that iterative extraction is valuable for pages with dense content, where a single pass may fail to capture all entities and relations.

## 5 CONCLUSION

In this paper, we proposed RECON, a graph-based RAG method that leverages MLLMs to automatically construct MMKGs from visually rich documents. By introducing adaptive intra-page reflection, RECON can adapt to different page complexities; by incorporating inter-page connection, it captures relations between entities across pages; and by constructing the QFVDS dataset with annotated answers and supporting facts, it enables reliable evaluation of document-level VQA. Through evaluations on multi-hop QA benchmarks and QFVDS dataset, RECON outperforms strong MM-RAG and graph-based baselines. Looking forward, combining RECON with Think-on-Graph (Ma et al., 2025) and query-driven MMKG construction (Bu et al., 2025) presents a promising direction for enabling deeper reasoning in MMRAG systems. These are worth studying as our future work.

## 6 ETHICS STATEMENT

This work does not involve human subjects, personally identifiable information, or sensitive data. All datasets employed in our experiments are publicly available and have been widely used in prior research, ensuring that our study adheres to standard ethical practices.

## 7 REPRODUCIBILITY STATEMENT

All datasets used in this work are publicly available, and our newly constructed QFVDS dataset will be released upon publication. Implementation details, including models, hyperparameters, and prompts, are provided in Section 4.3 and the Appendix. We include the core scripts in the supplementary material, and will release the full source code and dataset upon publication to ensure reproducibility.

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

|  | DLCV Slides | Picture Books | Env. Reports | World History |
|---|---|---|---|---|
| *Dataset size* | | | | |
| #Questions | 172 | 127 | 138 | 187 |
| *Per-question complexity (min / mean / max)* | | | | |
| Topics | 5 / 11.7 / 20 | 5 / 11.1 / 20 | 5 / 11.6 / 20 | 5 / 12.2 / 20 |
| Facts | 22 / 80.0 / 198 | 23 / 71.8 / 165 | 20 / 80.8 / 221 | 19 / 87.3 / 209 |
| Pages spanned | 16 / 70.1 / 174 | 15 / 51.8 / 109 | 11 / 40.0 / 106 | 13 / 51.9 / 113 |
| *Fact modality ratio (min / mean / max)* | | | | |
| Textual facts | 0.38 / 0.76 / 1.00 | 0.00 / 0.32 / 0.79 | 0.48 / 0.82 / 1.00 | 0.74 / 0.92 / 1.00 |
| Visual facts | 0.00 / 0.24 / 0.62 | 0.21 / 0.68 / 1.00 | 0.00 / 0.19 / 0.52 | 0.00 / 0.08 / 0.26 |

Table 4: Detailed QFVDS statistics across four domains. Each cell reports *min / mean / max*.

# A  APPENDIX

This appendix provides details on the QFVDS dataset, the computational cost of RECON, the use of LLMs, the limitations, and the prompts we used. The contents are organized into Appendices A.5, A.1, A.2, A.4, and A.3, respectively.

## A.1  QFVDS DATASET

We collected four diverse types of visually rich documents: (1) environmental reports from tech companies, (2) a world history textbook, (3) DLCV course slides on deep learning for computer vision, and (4) picture books. These datasets are derived from publicly available documents:

**Deep Learning for Computer Vision (DLCV) Slides.** Comprising 18 slide decks, this dataset[2] includes 1,984 pages, 2,018 figures, 75 tables, and 136,000 text tokens. The content is drawn from a deep learning and computer vision course, covering image classification, object detection, and societal impacts of AI.

**Environmental Reports.** Consisting of 5 corporate sustainability reports, this dataset includes 422 pages, 416 figures, 122 tables, and 229,000 text tokens. It documents environmental strategies from Google[3], Apple[4], Microsoft[5], Meta[6], and NVIDIA[7] (FY24 Sustainability Report), including goals for carbon reduction and renewable energy.

**World History Book.** A textbook[8] comprising 788 pages, 468 figures, 5 tables, and 442,000 text tokens. It traces global developments from prehistory to 1500 CE, covering early civilizations, empires, religious movements, and intercultural exchanges.

**Picture Books.** Collected from the Free Kids Books platform[9] under creative commons licensing, this dataset comprises 10 books totaling 247 pages, 240 figures, and fewer than 10,000 text tokens. Most books contain minimal text, relying heavily on visual storytelling through illustrations.

---

[2] https://cs231n.stanford.edu/slides/2024/
[3] https://sustainability.google/reports/google-2024-environmental-report/
[4] https://www.apple.com/environment/pdf/Apple_Environmental_Progress_Report_2024.pdf
[5] https://www.microsoft.com/en-us/corporate-responsibility/sustainability/report
[6] https://sustainability.atmeta.com/2024-sustainability-report/
[7] https://www.nvidia.com/en-us/sustainability/
[8] https://open.umn.edu/opentextbooks/textbooks/1418
[9] https://www.freekidsbooks.org/

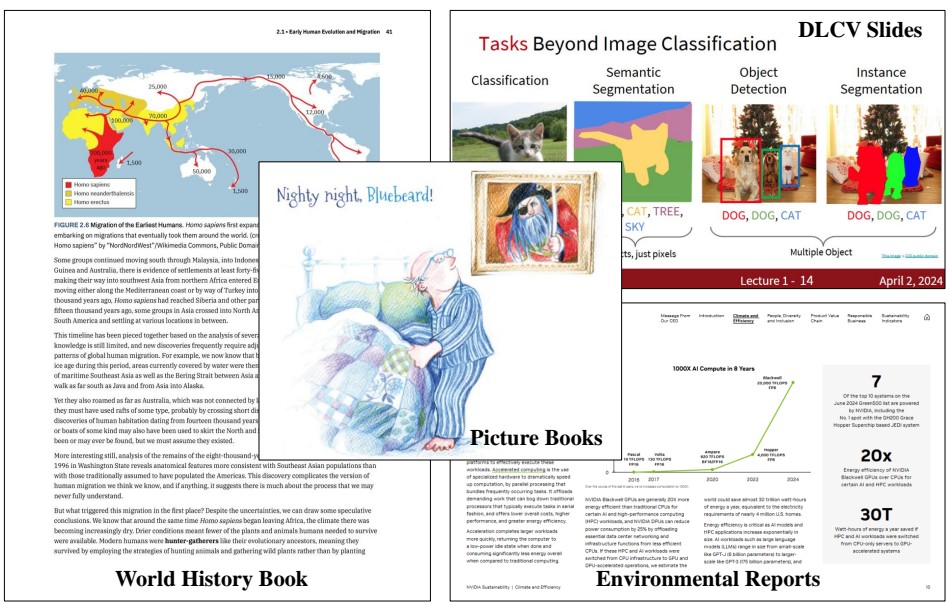

Figure 4: Representative pages from the four QFVDS domains: world history textbook, DLCV slides, environmental reports, and picture books. The examples highlight variation in layout and modality, ranging from dense text with tables/figures (slides, reports), narrative text with maps (history), to image-centric book (picture books).

### A.1.1 TOPICS AND FACTS

Table 4 reports detailed statistics of QFVDS across the four domains. Each question is associated with multiple topics and supported by textual and visual facts distributed across different pages. We report per-question *min / mean / max* for (i) number of topics, (ii) number of supporting facts, (iii) ratio of textual vs. visual evidence, and (iv) number of distinct supporting pages, i.e., the number of unique document pages on which the supporting facts are located.

DLCV Slides and Environmental Reports are dominated by textual evidence, with about 80 supporting facts per question on average. Picture Books contain fewer texts and rely more on visual evidence (mean ratio ≈0.68). World History requires the largest number of supporting facts (mean ≈87), mainly textual, spread across about 52 pages. These distributions show that QFVDS spans both text- and vision-centric settings and requires multi-page, multimodal reasoning.

### A.1.2 REPRESENTATIVE EXAMPLES

**Document snapshots.** Figure 4 shows representative pages from each domain. DLCV slides and environmental reports feature dense textual content, tables, and figures. World history combines long-form narrative text with maps and images. Picture books rely primarily on illustrations, with minimal textual context.

**QA Samples.** We present representative QA examples from the DLCV Slides (Table 5) and Picture Books (Table 6) domains. Each example consists of a question and its answer summary.

**Q**: How do the various components and techniques covered in the deep learning for computer vision course, such as softmax in attention mechanisms, gradient issues with saturated neurons, and bounding box regression loss, contribute to the understanding and development of neural networks, as illustrated in the course slides?

**ANS**:

**Activation Functions and Gradient Flow.** The course emphasizes the dual nature of activation functions like sigmoid and tanh: while sigmoid historically offers a **saturating "firing rate" interpretation** for neurons—providing a smooth curve that models how biological neurons respond—it also suffers from gradient issues. Specifically, **saturated neurons "kill" gradients**, meaning that when the input drives the activation to its asymptote, the derivative vanishes and impedes learning. This problem is shared by both sigmoid and tanh, as each can enter saturation regions where backpropagated gradients effectively become zero. Moreover, sigmoid's outputs are **not zero-centered**, which further slows convergence by introducing biases in gradient updates.

**Attention Mechanisms.** In the context of attention modules, the **softmax function** is the canonical choice for converting raw attention scores into a probability distribution over elements. Throughout the slides, it is reiterated that a **softmax function is applied to the attention weights** to ensure they sum to one and highlight the most relevant features. This normalization step—**softmax normalization**—is critical in letting the network weigh different spatial or channel locations appropriately when forming attended representations.

**Loss Functions and Objective Terms.** Two core loss formulations appear in the lecture material:

**Mean Squared Error (MSE) Loss.** For regression tasks (e.g., depth estimation or pixel-wise prediction), the loss is computed as: loss = np.square(y_pred - y).sum() This sums the squared differences between predictions `y_pred` and ground truth `y`.

**Bounding-Box Regression Loss.** In object detection frameworks like Faster R-CNN and Mask R-CNN, in addition to classification loss, there is a **bounding-box regression loss**. The slides depict both **classification loss** and **bounding-box regression loss** side by side, and note that **bounding box regressors are parameterized within the Faster R-CNN framework**, with the same idea extended to Mask R-CNN.

**Generative Models and Discriminators.** Generative Adversarial Networks (GANs) are illustrated through:

**Generated CIFAR-10 Samples:** Slides titled "Samples from the model" show the **left side containing digit samples** and the **right side featuring faces** labeled "Smiling woman," "Neutral woman," "Neutral man," and "Smiling Man."

**Discriminator Network:** A key component is the **discriminator network**, which **distinguishes between real and fake images**. It assesses whether a generated sample falls within the true data distribution, guiding the generator via adversarial loss signals.

**Depth Estimation.** For monocular or multi-view depth prediction, the network outputs a single-channel depth image with dimensions $1 \times H \times W$, where predictions can be viewed as $H \times W$. This standard output shape facilitates pixel-wise depth comparison using MSE loss.

**Convolutional Network Insights.** Two architectural considerations are highlighted:

**Receptive Field Size Calculation:** For a CNN with $L$ layers of kernel size $K$, each successive convolution **adds** $K - 1$ to the receptive field. Thus the total receptive field is:

$$1 + L\,(K - 1).$$

Moreover, each output element depends on a $K \times K$ patch in the input.

**Visualizing Filters:** The course agenda explicitly includes **visualizing filters**, which helps students understand what each convolutional kernel has learned and how feature hierarchies emerge.

**Course Structure and Credits.** The slides carry clear metadata and attribution:

**Lecture Labeling:** Although the material is consistently titled "Lecture 3," the slides also note a **lecture number of 10** in certain contexts.

**Lecture Dates:** Two dates appear in the sequence of lectures—**April 18, 2024**, and **May 23, 2024**.

**Credit:** All slides are **credited to Justin Johnson**, acknowledging the source of the presented content.

This cohesive overview shows how elements ranging from activation design and attention normalization, through loss formulations and adversarial training, to optimization strategies and architectural analysis, combine to deepen understanding and drive the development of robust computer vision models.

Table 5: QA example from the DLCV subset of QFVDS.

**Q**: How do the various characters and elements in the picture books, such as the Billy Goats Gruff, Lwando, Cheta the Snail, and the crocodile, interact with their environments and contribute to the themes of adventure, creativity?

**ANS**:

**Adventure and the Journey Motif.** Across these picture books, characters undertake physical and emotional journeys that underscore themes of adventure. In the familiar retelling of *The Three Billy Goats Gruff*, the goats' "TRIP-TRAP! TRIP-TRAP! TRIP-TRAP!" crossing of the creaking, groaning bridge evokes suspense as each goat answers the troll's roar, "WHO'S THAT TRIP-TRAPPING OVER MY BRIDGE?" Once across, the Middle Billy Goat Gruff and then the Big Billy Goat Gruff ascend a hillside blanketed in thick, green grass where they joyfully reunite. Similarly, Cheta the Snail—purple-bodied with a yellow spiral shell and a blue scarf—expresses a heartfelt wish: "I want to ride on the bus like you." Cheta's desire to take a bus journey and travel the highway conveys both the excitement and the vulnerability of setting out into the bigger world. Even the underwater realm hosts a small adventure: several orange, pink, blue, and yellow fish swim in a blue area at the bottom of the page, suggesting that exploration takes many forms and scales.

**Creativity, Play, and Self-Expression.** Creativity springs to life when children and animals improvise with everyday objects. Lwando dons a cardboard box as a costume—box firmly on his head—paired with his blue shirt and shorts. Holding hands with his friend Oyiso, Lwando transforms simple materials into imaginative play, demonstrating how ingenuity can turn an ordinary scene (complete with paved ground featuring a light-colored, striped pattern) into an immersive adventure. Another boy likewise wears a cardboard box on his head, reinforcing the theme of the box's celebration of playful invention. In *How to Tame a Monster*, readers learn that "You can only tame monsters by helping them feel safe." This idea broadens creativity into emotional intelligence, teaching that imaginative concepts (monsters) can be guided by kindness. Elsewhere, rabbits sporting red clothing spark joyful action: a large rabbit in a red tank top clasps its face in delight while two smaller rabbits raise their arms, clad in red shorts; two more wear red jerseys—one emblazoned with the number 1—and another perches on a track to put on a shoe. Such scenes encourage readers to invent games, sports, and performances.

**Everyday Acts and Making a Difference.** The power of small deeds to shape lives resonates through simple, solitary accomplishments. One story shows a hen meticulously stirring a bowl with a wooden spoon, kneading dough and baking bread all by herself—an ode to self-reliance and the quiet pride of creation. The phrase "Make a Difference," often accompanied by #FKBMakeaDifference, reminds us that books—and by extension small narratives—hold the capacity to change minds and hearts. A man in glasses, a red shirt, and an apron attends to his own task—perhaps baking, cooking, or crafting—demonstrating how personal dedication, however humble, enriches community. Nearby, a crocodile with a green body, striped shirt, and pink shoes holds both a large orange shoe and a small snack, blending fun and function, and illustrating that even creatures can contribute helpful acts or share provisions.

**Setting the Stage: Environments and Visual Storytelling.** Each illustration is anchored by rich backgrounds and carefully noted details. Pages numbered 6/15, 13, 14/17, 15/17, 21, 25, 37, and even 64 remind readers of the book's structure; notably, page 21 features six pictures, allowing multiple vignettes to unfold simultaneously. In many scenes, green trees, bushes, and rolling hills frame the action, while clusters of two or more houses nestle among them, creating a sense of community. One paved area, striped and light-colored, signals a roadway or plaza—perfect for Lwando's make-believe or Cheta's bus stop. Dogs appear in this world too: a white dog sits calmly on the grass, another walks purposefully, and a third lies down, evoking both companionship and relaxation. Overhead, a lion with a large mane charges on two legs toward another character, injecting a dash of thrilling motion and reminding readers that even in pastoral settings, the unexpected can happen.

**Interwoven Themes: Unity Through Exploration and Kindness.** Taken together, these characters and elements highlight that adventure comes in many guises—from the perilous bridge of the Billy Goats Gruff to the humble kitchen where a hen bakes alone. Creativity blooms when children like Lwando and the box-wielding boy repurpose ordinary items, while lessons from *How to Tame a Monster* and the hashtag #FKBMakeaDifference underscore the transformative power of empathy and books themselves. Whether racing across striped pavements or drifting beneath the waves with colorful fish, each scene contributes to a tapestry in which exploration, imagination, and the desire to make a difference coalesce into stories that inspire readers of all ages.

Table 6: QA example from the Picture Books subset of QFVDS.

## A.2 COMPUTATIONAL COST

**MMKG Construction.** RECON constructs MMKGs directly from document images, which requires more computation than text-only methods (Gutiérrez et al., 2025; Guo et al., 2024; Edge et al., 2024). The dominant cost lies in MLLM-based entity extraction and reflection, and runtime increases with both document length and page complexity.

For smaller collections such as Picture Books (247 pages, 281 figures), construction was completed in about 26.7 minutes using the *GPT-4o-mini* API. This time included roughly 13 minutes for entity extraction and reflection, 10 minutes for MLLM-based cross-page relation filtering, and 3 minutes for GME indexing. Larger and more complex documents, such as Environmental Reports (422 pages, 416 figures, 122 tables), required approximately 94 minutes in total: 52 minutes for extraction and reflection, 30 minutes for cross-page relation filtering, and 12 minutes for GME indexing. All GME indexing was conducted on a single RTX 5090 GPU (32GB memory).

Overall, these results show that RECON scales predictably with document size and complexity while still offering a practical and cost-effective alternative to manual annotation. Once constructed, MMKGs can be reused across multiple queries without additional overhead, making the one-time cost a worthwhile investment for scalable multimodal reasoning.

**Retrieval and Answer Generation.** We benchmarked retrieval and answer generation on Environmental Reports, using the *GPT-4o-mini* API with a single RTX 5090 GPU (32GB). At query time, RECON executes MMKG (entity-relation) retrieval and page-level retrieval in parallel. Retrieval is efficient, averaging about 1.0s in total ($\approx$1.0s for MMKG and $\approx$0.4s for pages). Answer generation is more demanding: RECON first produces candidate answers from both the MMKG ($\approx$20.5s) and pages ($\approx$26s) in parallel, followed by a second-stage fusion step ($\approx$16s) to consolidate them. This results in an end-to-end latency of about 42s.

In comparison, GME (MMRAG) completes page retrieval ($\approx$0.4s) and answer generation ($\approx$26s) in $\approx$26.4s total. While RECON incurs higher latency, it provides MMKG-augmented answers that integrate structured and unstructured evidence, enabling stronger reasoning capabilities.

## A.3 THE USE OF LARGE LANGUAGE MODELS

We employed LLMs at several stages of our work. *GPT-4o-mini* was used for graph construction, including both intra-page reflection and inter-page connection. The *o4-mini* model was used for evaluation and dataset construction, generating candidate questions, answers, and supporting facts that were subsequently validated through manual review. All LLM usage in this work was conducted in a zero-shot setting without fine-tuning, ensuring both accessibility and reproducibility. LLMs were not used for research ideation or paper writing beyond minor language polishing.

## A.4 LIMITATIONS

While RECON demonstrates strong performance, it also has several limitations. First, its reasoning ability remains limited: the current retrieval operates mainly on dense retrieval (for MMKG and document pages). A promising direction is to integrate RECON with reasoning-oriented retrieval strategies such as Think-on-Graph (Ma et al., 2025) or with dynamically constructed MMKGs (Bu et al., 2025) at query time, which could enhance flexibility. However, ensuring that such integrations remain efficient poses an open challenge for future exploration. Second, RECON currently supports only textual and visual modalities. Real-world scenarios often involve richer signals such as audio, video, or interactive content. Extending MMKG construction to incorporate these modalities would further broaden the scope of multimodal reasoning. These are worth studying as our future work.

## A.5 MLLM PROMPTS

For reproducibility, we provide all prompts used in our pipeline. These include prompts for entity-relation extraction and reflection (Prompts 1, 2), inter-page connection filtering (Prompt 3), and MMKG-augmented answer generation (Prompts 4, 5, 6). We also include prompts for fact extraction and QFVDS QA generation (Prompts 7, 8), as well as evaluation prompts for multi-hop QA and pairwise comparison in QFVDS (Prompts 9, 10).

### Prompt 1: Entity Relation Extractor

```
1   Given a single page image, along with the context about this document,
        entity types, identify all important entities and relationships
        in JSON format.
2   Use {language} as output language.
3
4   ---Instructions---
5   1. Process the Input:
6      - The input is a single page image.
7      - From this image, capture:
8        - All the textual entities derived from the text content of the
            page image.
9        - All the visual entities derived from figures, tables, or other
            visual components.
10
11  2. Entity Extraction:
12     For each identified entity, extract:
13     - "type": Always "ent"
14     - "en": Name in the same language as the input text
15     - "et": One of: [{entity_types}]
16     - "ed":
17        - If entity is from text: Comprehensive description of the
            entity's attributes and activities.
18        - If entity is from a visual component: Description should
            focus on visual characteristics (appearance, layout,
            composition, style, notable visual features).
19     - "es": "text" if derived from text content, "visual" if derived
            from a visual component.
20
21  3. Relationship Extraction:
22     - Extract relationships not only between text entities, but also
            between text entities and visual entities, as well as between
            visual entities themselves.
23     - For each pair of related entities:
24        - "type": Always "rel"
25        - "se": Name of the source entity
26        - "te": Name of the target entity
27        - "rd": Explanation of the relationship (e.g., a logo
                representing a company, a diagram illustrating a process
                described in text).
28        - "rk": Array of high-level keywords summarizing the relationship
29        - "rs": Numeric score for strength of the relationship
30
31  4. Content Keywords:
32     - Add one object with:
33        - "type": "content_keywords"
34        - "keywords": Array of overarching themes or concepts
35
36  5. Output Rules:
37     - Return a single JSON array containing all entity, relationship,
            and content keyword objects.
38     - Output must be valid JSON, no extra commentary or formatting.
39     - Output only in {language}.
40
41  ---Examples---
42  {examples}
```

```
43
44 ---Metadata about the document---
45 Entity types:
46 {entity_types}
47
48 Total Document Pages: {total_page}, Now on Page: {now_page}
49
50 Output (JSON array only):
```

### Prompt 2: Entity Relation Reflector

```
1 You are an entity-relation reflector. Given a single page image, along
       with the entities, and relations extracted before, analyze
     entities and their existing relations, then propose missing
     entities and relation types.
2
3 Return your answer in JSON only. Follow the JSON schema exactly:
4 {
5     "potential_missing_entities": string[],
6     "potential_missing_relations": [
7         {
8             "entity": string,
9             "missing_relations": string[]
10        }
11    ],
12    "done_with_this_page": {
13        "reason": string,
14        "done_extract": boolean
15    }
16 }
17
18 PAGE-GROUNDED EVIDENCE ONLY:
19 - Suggestions must be supported by this page image: text, caption,
       label, clearly visible figure/diagram/table/icon.
20
21 DONE WITH THIS PAGE:
22 - Provide a short reason.
23 - Set done_extract = true if remaining critical links (if any) likely
       require other pages; otherwise false.
24
25 Output (JSON array only):
```

### Prompt 3: MLLM Filter for Inter-Page Connection

```
1 You are a multimodal relation filter. Decide whether the CANDIDATE
       entities are actually connected to the SOURCE entity.
2 Use only the provided text; do not assume facts not stated here. If
       uncertain, do not link.
3
4 DECISION PRINCIPLES
5 - Evidence must be explicit in the provided descriptions.
6 - Name overlap alone is insufficient unless the description explicitly
        asserts they are the same or related entities.
7
8 MINIMUM EVIDENCE TO LINK (must satisfy at least one)
9 - Direct statement of the relationship between SOURCE and candidate.
10 - Explicit cross-mention (SOURCE mentioned in candidate description or
        vice versa) describing the relation.
11
12 REJECTION TRIGGERS
13 - Only similarity or category overlap is present.
14 - Only partial or fuzzy name match, unresolved acronym.
15 - Relationship requires outside knowledge or unstated assumptions.
16 - The text is ambiguous, speculative ("may", "might", "possibly"), or
        lacks a concrete verb indicating a relation.
```

```
17
18  OUTPUT FILTER
19  - Compute a confidence score (rs) from 0-10 based solely on the
        provided text.
20  - Include a candidate in the output ONLY if rs is greater than 7 and
        the rationale can quote or paraphrase the specific evidence phrase
        (s).
21  - If NO candidate fits, output an empty list: [].
22
23  RATIONALE REQUIREMENTS
24  - 1-2 sentences, referencing the exact supporting phrase(s) from the
        given descriptions using short quotes where possible.
25  - If applicable, state whether the link is textual or visual (e.g., "
        caption shows", "label reads").
26
27  OUTPUT FORMAT (exact)
28  - Output a JSON list. Each element is a dict with exactly:
29  {{
30      "type": "rel",
31      "se": "source entity",
32      "te": "target candidate entity",
33      "rd": "1-2 sentence rationale referencing provided descriptions;
            mention textual or visual if relevant",
34      "rk": ["2-5 short keywords", "..."],
35      "rs": integer 0-10 confidence score
36  }}
37  - No extra text, no markdown, no trailing commas, no additional fields
        .
38
39  INPUT
40  ---- Entity Query ----
41  {QUERY}
42
43  ---- Source Entity ----
44  entity: "{SOURCE_ENTITY}"
45  description: {SOURCE_DESCRIPTION}
46
47  ---- Candidate Entities ----
48  {CANDIDATES}
49
50  Output (JSON array only):
```

Prompt 4: MMKG-augmented Answer Generation (MMKG part)

```
1   ---Role---
2
3   You are a professional assistant responsible for answering questions
        based on a knowledge graph.
4
5   ---Goal---
6
7   Generate a thorough, detailed, and complete answer to the query that
        incorporates all relevant information from the knowledge graph. Do
         not simplify or summarize aggressively-aim for maximum coverage,
        including different perspectives, detailed facts, and nuanced
        insights from both sources. If you don't know the answer, just say
         so. Do not make anything up or include information where the
        supporting evidence is not provided.
8
9   When handling relationships with timestamps:
10  1. Each relationship has a "created_at" timestamp indicating when we
        acquired this knowledge
11  2. When encountering conflicting relationships, consider both the
        semantic content and the timestamp
```

```
12  3. Don't automatically prefer the most recently created relationships
        - use judgment based on the context
13  4. For time-specific queries, prioritize temporal information in the
        content before considering creation timestamps
14
15  ---Knowledge Graph---
16  {context_data}
17
18  ---Response Rules---
19
20  - Use markdown formatting with appropriate section headings
21  - Please respond in the same language as the user's question.
22  - List up to 5 most important reference sources at the end under "
        References" section. Clearly indicating whether each source is
        from Knowledge Graph (KG), and include the file path if available,
         in the following format: [KG] file_path
23  - If you don't know the answer, just say so.
24  - Do not make anything up. Do not include information not provided by
        the Knowledge Base.
25
26  Response:
```

Prompt 5: MMKG-augmented Answer Generation (Document Images part)

```
1   ---Role---
2
3   You are a helpful assistant responding to user query about Document
        Images provided below.
4
5   ---Goal---
6
7   Generate a thorough, detailed, and complete answer to the query that
        incorporates all relevant information from the Document Images. Do
         not simplify or summarize aggressively-aim for maximum coverage,
        including different perspectives, detailed facts, and nuanced
        insights from both sources. If you don't know the answer, just say
         so. Do not make anything up or include information where the
        supporting evidence is not provided.
8
9   When handling content with timestamps:
10  1. Each piece of content has a "created_at" timestamp indicating when
        we acquired this knowledge
11  2. When encountering conflicting information, consider both the
        content and the timestamp
12  3. Don't automatically prefer the most recent content - use judgment
        based on the context
13  4. For time-specific queries, prioritize temporal information in the
        content before considering creation timestamps
14
15  {content_data}
16
17  ---Response Rules---
18
19  - Use markdown formatting with appropriate section headings
20  - Please respond in the same language as the user's question.
21  - List up to 5 most important reference sources at the end under "
        References" section. Clearly indicating each source from Page
        Images(PI), and include the file path if available, in the
        following format: [PI] image_path
22  - If you don't know the answer, just say so.
23  - Do not include information not provided by the Document Chunks.
24  - Addtional user prompt: {user_prompt}
25
26  Response:
```

**Prompt 6: MMKG-augmented Answer Generation (Final two-stage combination)**

```
1   ---Role---
2
3   You are a professional assistant responsible for answering questions
        based on both a knowledge graph and visual information extracted
        from document images containing relevant textual and visual
        content (e.g., scanned pages, slides, charts, or forms).
4
5   You are provided with a user query and two independent answers:
6   1. An answer based on the knowledge graph.
7   2. An answer based on the document images.
8
9   Your task is to analyze the user's query and integrate the two
        provided answers into a single comprehensive response. Do not omit
         any relevant points from either source. When the answers conflict
         or provide complementary insights, use grounded reasoning to
        reconcile them. If the knowledge graph provides explicit facts, do
         not override them unless contradicted by strong visual evidence.
10
11  ---Query---
12
13  {query}
14
15  ---Input Answers---
16
17  - Answer from Knowledge Graph:
18  {kg_answer}
19
20  - Answer from Document Images:
21  {image_answer}
22
23  ---Goal---
24
25  Generate a thorough, detailed, and complete answer to the query that
        incorporates all relevant information from both Answers from the
        Knowledge Graph and the Document Images. Do not simplify or
        summarize aggressively; aim for maximum coverage, including
        different perspectives, detailed facts, and nuanced insights from
        both sources. If you don't know the answer, just say so. Do not
        make anything up or include information where the supporting
        evidence is not provided.
26
27  When handling information with timestamps:
28  1. Each piece of information (both relationships and content) has a "
        created_at" timestamp indicating when we acquired this knowledge.
29  2. When encountering conflicting information, consider both the
        content/relationship and the timestamp.
30  3. Don't automatically prefer the most recent information - use
        judgment based on the context.
31  4. For time-specific queries, prioritize temporal information in the
        content before considering creation timestamps.
32
33  ---Response Rules---
34
35  - Generate a final answer that integrates both inputs.
36  - Use markdown formatting with appropriate section headings.
37  - Organize answer in sections focusing on one main point or aspect of
        the answer
38  - List up to 5 most important reference sources at the end under "
        References" section. Clearly indicating whether each source is
        from Knowledge Graph (KG) or Page Images (PI), and including the
        file path if available, in the following format: [KG/PI] file_path
39  - Ensure the response maintains continuity with the conversation
        history.
```

```
40  - If you don't know the answer, just say so. Do not make anything up.
41  - Do not include information not provided by the inputs.
```

### Prompt 7: Atomic Facts Extraction

```
1   You extract concise, verifiable FACTS from a single document page
        image.
2
3   STRICT requirements:
4   - Output a JSON object with exactly one key: "facts": {{ "fact":
        string, "source": "text" | "figure" }}[].
5   - Produce atomic facts when possible; fewer only if the page has
        little content.
6   - Each fact MUST be a single self-contained declarative sentence with
        its subject and any units/dates, and MUST end with a period.
7   - Avoid deictic/layout words and vague pronouns: do NOT use "this", "
        that", "these", "those", "above", "below", "left", "right", "the
        following", "see figure/table".
8   - A fact is "text" if it comes from paragraphs, headings, lists,
        tables, or captions.
9   - A fact is "figure" if it comes from a chart/diagram/photo/graphic,
        including labels/legends/axes inside the graphic.
10  - Do not invent information; only state what appears on the page.
11  - If the page has no legible content, return {{ "facts": [] }}.
12
13  Extract self-contained facts present on this page.
14  For each fact, also label its source as "text" or "figure" using the
        rules above.
15
16  Return ONLY JSON like:
17  {{"facts":[{{"fact":"...", "source":"text"}}, {{"fact":"...", "source"
        :"figure"}}]}}
```

### Prompt 8: QFVDS QA Pairs Generation

```
1   You are a "Query-Focused Summary (QFS) QA pair generator"
2
3   ---Goal---
4
5   I will provide you with "{topic_length} topics" along with their
        corresponding "topic descriptions and facts" (these are extracted
        from a source document {document_name}).
6   Generate a question along with a thorough, detailed, and complete
        answer that incorporates all relevant information from the Provide
         Topics and Facts.
7   Do not simplify or summarize aggressively-aim for maximum coverage,
        including different perspectives, detailed facts, and nuanced
        insights.
8   Do not make anything up or include information where the supporting
        evidence is not provided.
9
10  ---Topics and Facts---
11  {content_data}
12
13  ---Response Rules---
14
15  - Target format and length: Multiple Paragraphs
16  - Use markdown formatting with appropriate section headings
17  - Concisely cover all {topic_length} topics in a coherent and
        logically organized manner.
18  - Preserve factual accuracy without adding external information.
19
20  Response:
21  "Output in JSON format (do NOT add anything else)":
```

```
22   {{
23       "question": ...,
24       "answer": ...
25   }}
```

### Prompt 9: Multi-hop QA Evaluation

```
1    Compare the "REFERENCE ANSWER" with the "CANDIDATE" answer and judge
2    whether they express the *same factual content*.
3    Ignore writing style, order, or extra background; focus only on
         whether
4    the key facts (names, dates, amounts, percentages, etc.) truly match.
5
6    Instructions
7    ------------
8    1. If every critical fact in the candidate unambiguously agrees with
         the
9       reference, label "correct".
10      - Paraphrases or synonymous wording are acceptable.
11      - Numbers must be equal after unit conversion/rounding (e.g. "431 M
           " = "$431 million").
12   2. Otherwise, label "incorrect" (do *not* use "partial", "unknown",
         etc.).
13
14   Output format
15   -------------
16   Return a single-line JSON object *exactly* like:
17
18   {"label": "correct", "rationale": "..."}
19
20   - "label" – either "correct" or "incorrect" (lowercase).
21   - "rationale" – a clear, factual explanation in "under 50 words".
22     - If correct, explain briefly why the key facts match.
23     - If incorrect, identify the first major contradiction (e.g.
           different person, date, amount).
24     - Use specific, content-based justifications (not generic statements
           ).
25
26   Constraints
27   -----------
28   * Do not include extra keys, arrays, markdown, or formatting.
29   * Do not quote the input or repeat the entire answer.
30   * Focus only on factual correctness – avoid style or completeness
         judgments.
31   * Do not make up metrics or mention unrelated information like
         percentages unless directly relevant.
```

### Prompt 10: QFVDS Pairwise Evaluation

```
1    You will evaluate two answers to the same question using two criteria:
          "SemanticSimilarity" and "Faithfulness".
2    Treat the "GT summary" as the reference narrative and the "GT facts"
         as the canonical checklist.
3
4    Definitions and rubric:
5
6    - "SemanticSimilarity": How closely does the answer's meaning align
         with the GT summary?
7
8    - "Faithfulness": No contradictions with GT facts or the GT summary,
         and enumerates all GT facts to assess how many are clearly present
         .
9
10   Instructions:
```

```
11  1) Base judgments "only" on the Question, GT summary, and GT facts.
12  2) In explanations, "cite fact indices" like [F2], [F4] when referring
         to specific GT facts, and note coverage (e.g., "Covered: F1,F3,F4
         ; Missing: F2").
13  3) For each criterion, pick a winner ("Answer 1" or "Answer 2") and
         explain why, referencing fact indices where relevant.
14
15  Here are the GT facts (treat as canonical; enumerate internally as [F1
         ]..[Fn]):
16  {gt_facts}
17
18  Here is the question:
19  {query}
20
21  Here is the ground truth (GT) answer summary:
22  {gt_answer}
23
24  Here are the two answers:
25
26  "Answer 1 Start"
27  {answer1}
28  "Answer 1 End"
29
30  "Answer 2 Start"
31  {answer2}
32  "Answer 2 End"
33
34  Evaluate both answers using the two criteria above and provide
         detailed explanations for each criterion. In explanations,
         reference GT facts by index (e.g., [F3]) whenever applicable.
35
36  Output your evaluation in the following JSON format (do not include
         additional fields):
37
38  {{
39      "SemanticSimilarity": {{
40          "Winner": "[Answer 1 or Answer 2]",
41          "Explanation": "[Explain which answer more closely matches the
                   GT meaning. Refer to specific GT phrases; cite facts like
                   [F1], [F2] when helpful.]"
42      }},
43      "Faithfulness": {{
44          "Winner": "[Answer 1 or Answer 2]",
45          "Explanation": "[Report any "material" contradictions (if any)
                   and compare "coverage" (e.g., Covered: F1,F3; Missing: F2
                   ,F4). Cite facts such as [F2], [F5].]"
46      }}
47  }}
```

