# OpenReview forum: "RECON: Multimodal GraphRAG for Visually Rich Documents with Intra-Page Reflection and Inter-Page Connection"
_ICLR.cc/2026/Conference — ICLR 2026 Conference Withdrawn Submission_

### Official Review · Reviewer_XiiA · 2025-10-30

**Soundness:** 2
**Presentation:** 2
**Contribution:** 2
**Rating:** 4
**Confidence:** 5

**Summary:**

Current methods like Multimodal Retrieval-Augmented Generation (MMRAG) and text-only graph approaches struggle with comprehending long, visually rich documents, as MMRAG often misses holistic context and existing knowledge graphs are not built automatically for multimodal content.
RECON overcomes these limitations by automatically constructing a Multimodal Knowledge Graph (MMKG) that captures both textual and visual information to provide a structured, global understanding of the entire document.
Its process involves two main stages: "Intra-page Reflection" to adaptively extract entities and relations within each page, followed by "Inter-page Connection" to link these individual graphs into a single coherent MMKG.

**Strengths:**

1. The paper introduces a novel two-stage method for automatically building MMKGs from visually rich documents. The first stage, "Intra-page Reflection," adaptively extracts entities and relations within each page based on content complexity. The second stage, "Inter-page Connection," merges these individual page graphs and establishes links between entities across different pages to form a cohesive, document-level knowledge graph. This automated, zero-shot process is designed to overcome the limitations of manual graph creation or computationally intensive query-time construction found in previous works.
2. The work addresses a gap in model evaluation by developing the Query-Focused Visual Document Summaries (QFVDS) dataset. This new benchmark is specifically designed to test long-range reasoning in visual documents and is distinctive for providing annotated answers along with the corresponding supporting facts necessary to derive them. Spanning four varied document types—including corporate reports, a textbook, lecture slides, and picture books—QFVDS enables a more direct and reliable assessment of factual correctness and semantic similarity for document-level VQA tasks.

**Weaknesses:**

1. The claim that RECON is superior in text-only benchmarks is questioned, as its average performance is nearly identical to the strongest baseline. The reviewer suggests a significance test or more cautious wording.
2. The paper acknowledges its reliance on dense retrieval rather than more advanced graph reasoning strategies. This, combined with higher end-to-end latency compared to strong retrieval baselines, raises questions about its production readiness without further optimization.
3. The QFVDS dataset's construction, which uses an MLLM to generate questions and answers that are then filtered by humans, may introduce bias. The reviewer notes the lack of a control experiment against a purely human-annotated benchmark.
4. The paper notes that constructing the Multimodal Knowledge Graph (MMKG) for a 422-page document took approximately 94 minutes. While presented as practical, this could become a significant bottleneck for much larger documents or large-scale document collections, which are common in enterprise settings. The cost of using a powerful model like GPT-4o-mini for iterative extraction on thousands or millions of documents could also be substantial.
5. RECON's pipeline is multi-staged and complex, involving parallel MLLM calls, a separate reflector step, entity merging, similarity calculations for inter-page connection, and a two-stage answer generation process. This complexity can make the system difficult to implement, debug, and maintain in a real-world application compared to more straightforward end-to-end RAG models.
6. The paper uses a "knee detection method" to automatically determine the threshold for identifying low-degree entities for inter-page connection. This is an interesting heuristic, but its stability and optimality are not deeply explored. The performance could be sensitive to this threshold, and it's unclear how robust this method is across different graph structures generated from various documents.

[Minor]
In the answer generation prompts, the model is instructed to use judgment when encountering conflicting information, considering timestamps. However, the paper doesn't provide a systematic evaluation of how well the model performs this reconciliation. In complex, real-world documents, conflicting data is a common problem, and simply instructing the model to "use judgment" may not be a sufficiently robust solution.

**Questions:**

Please refer to Weakness section.

---

### Official Review · Reviewer_mzMH · 2025-10-31

**Soundness:** 3
**Presentation:** 2
**Contribution:** 2
**Rating:** 4
**Confidence:** 2

**Summary:**

This paper proposes RECON, a Multimodal Graph-based RAG that builds MMKGs in two stages: 1) Intra-page REflection (extracts/integrates page-level textual-visual relations) and 2) Inter-page CONnection (links cross-page relations into a global graph). It also constructs a QFVDS dataset (with annotated answers/facts) to fill the evaluation gap. Experiments show RECON outperforms existing MMRAG on VQA datasets and QFVDS.

**Strengths:**

1) The proposed method of automatically constructing MMKGs for VRD is a further step to better understanding long documents, and the proposed two-stage framework, intra-page reflection and inter-page connection, are effective modules to implement this automation.
2) They curated a new dataset, QFVDS, to evaluate the proposed RECON, and both answers and supporting facts annotations are supplied.
3) Experimental results show the superiority of the proposed method.

**Weaknesses:**

1) The entity only includes text and figure, which means that table is treated as whole figure that will lead to missing of fine-grained information.
2) Tab. 1, in text-only setting, RECON achieves marginal improvement to HippoRAG2 (66.93 VS 66.87), please explain it in detail.
3) The abbreviation definition is confusing, and the main body should redefine those defined in abstract.

**Questions:**

See Weaknesses.

---

### Official Review · Reviewer_EcJH · 2025-11-01

**Soundness:** 3
**Presentation:** 2
**Contribution:** 3
**Rating:** 4
**Confidence:** 3

**Summary:**

This paper introduces RECON, a multimodal GraphRAG framework for comprehending long, visually-rich documents. The authors address the challenge of document-level reasoning by proposing a method to automatically construct multimodal knowledge graphs (MMKGs). RECON uses a zero-shot, two-stage pipeline: (1) an "Intra-page REflection" process to iteratively extract textual and visual entities/relations within each page, and (2) an "Inter-page CONnection" stage to merge page-level graphs and filter cross-page links. The paper also contributes QFVDS, a new benchmark dataset for evaluating query-focused, long-range VQA. Experimental results show that RECON outperforms MMRAG and graph-based RAG baselines on existing multi-hop QA datasets and the QFVDS benchmark.

**Strengths:**

1. Addresses a Key Challenge: RECON specifically targets the holistic comprehension of long, visually-rich documents, a notable weakness of existing MLLM and RAG methods.
2. Contribution of a New Benchmark: The paper introduces the QFVDS dataset, a dedicated benchmark for evaluating long-range, cross-page, multimodal question answering.
3. Strong Performance of a Training-Free Framework: As a zero-shot framework, RECON demonstrates strong performance, outperforming existing MMRAG and GraphRAG baselines on multi-hop QA, multimodal QA, and the new QFVDS benchmark.

**Weaknesses:**

1. The method's novelty is primarily concentrated on the knowledge graph construction stage, with insufficient novelty in the RAG pipeline itself.
2. The description of the RAG process is not sufficiently detailed.
3. Regarding experimental evaluation, the assessment is limited to multi-hop QA and multimodal QA. There appears to be no evaluation on existing document understanding benchmarks, such as DocVQA or MMLONGBENCH-DOC.
4. The paper lacks sufficient visualization.
5. There is a lack of failure case analysis, which makes it difficult to understand the main challenges of the current task.

**Questions:**

What is the main difference between the QFVDS (query-focused visual document summaries) task and the QFS task from GraphRAG? Is QFVDS simply a direct extension of QFS into the visual document domain?

---

### Official Review · Reviewer_RWzs · 2025-11-01

**Soundness:** 3
**Presentation:** 3
**Contribution:** 2
**Rating:** 4
**Confidence:** 4

**Summary:**

This paper introduces a multimodal knowledge graph construction framework designed to enhance intra- and inter-page understanding using LLMs for retrieval-augmented, multi-hop document VQA. To better evaluate the proposed method, the authors present a new dataset tailored for multi-hop, cross-page document-level QA. The framework demonstrates promising performance across both text-only and multimodal scenarios. While the overall workflow appears reasonable and well-structured, several concerns arise regarding the methodological novelty and the clarity of the data flow during the inference stage.

**Strengths:**

1. The research aim and direction is quite meaningful which highlights the bottlenect in the area. And the proposed method could mitigate the research gaps effectively based on the provided experimental results.
2. The proposed method looks workable and evaluated on varoius domains to demonstrate the robustness.
3. Reaonable performance improving achieved after compared with various baselines.

**Weaknesses:**

The main issue lies in the method section, particularly regarding the novelty and technical depth of the proposed framework for KG construction and multimodal reasoning.

- Novelty and Research Depth
The framework mainly relies on prompt design for KG construction and retrieval. While exploring intra- and inter-page correlations is a meaningful direction, the work lacks deeper analysis or methodological innovation beyond prompt engineering, limiting its research depth.

- Representation Limitations
The document content appears to be represented only through text or dense multimodal embeddings. This design neglects fine-grained spatial and structural details that are crucial for accurate document understanding.

- Parsing Robustness
The document parsing pipeline depends solely on MLLMs. For noisy, handwritten, or scanned documents, such reliance could significantly degrade performance. Even for digital-born documents, MLLMs struggle with long inputs and logical correlations. Integrating specialized document parsing agents or preprocessing modules would be beneficial.

- Error Propagation and Bias Control
If the framework relies on a single LLM during KG construction, error propagation and bias may become significant. If multiple LLMs are used, it is unclear how their outputs are balanced or reconciled to ensure consistency and reliability.

- Answer Generation Strategy
The answer generation component lacks depth and appears to simply combine outputs. More sophisticated reasoning, consistency checking, or multi-hop QA mechanisms should be introduced to enhance the quality of generated answers.

**Questions:**

- How are fine-grained visual and structural details (e.g., layout hierarchy, object spatial relations) preserved or modeled in your multimodal representation?
- Since the framework depends on MLLMs for parsing, how does it handle noisy, handwritten, or scanned documents where layout recognition is unreliable?
- How do you mitigate error propagation when the same LLM is used for both parsing and KG construction?
- What is the computational cost of your framework during KG construction and answer generation, considering multiple LLM calls and multimodal processing?

---

### Note · Authors · 2025-11-14

I have read and agree with the venue's withdrawal policy on behalf of myself and my co-authors.